# Accumbens cholinergic interneurons dynamically promote dopamine release and enable motivation

Ali Mohebi[1], Val L Collins[1], Joshua D Berke[1,2,3,4]*

[1]Department of Neurology, University of California, San Francisco, San Francisco, United States; [2]Department of Psychiatry and Behavioral Sciences, University of California, San Francisco, San Francisco, United States; [3]Neuroscience Graduate Program, University of California, San Francisco, San Francisco, United States; [4]Weill Institute for Neurosciences, University of California, San Francisco, San Francisco, United States

*For correspondence:
joshua.berke@ucsf.edu

Competing interest: The authors declare that no competing interests exist.

**Abstract.** Motivation to work for potential rewards is critically dependent on dopamine (DA) in the nucleus accumbens (NAc). DA release from NAc axons can be controlled by at least two distinct mechanisms: (1) action potentials propagating from DA cell bodies in the ventral tegmental area (VTA), and (2) activation of β2* nicotinic receptors by local cholinergic interneurons (CINs). How CIN activity contributes to NAc DA dynamics in behaving animals is not well understood. We monitored DA release in the NAc Core of awake, unrestrained rats using the DA sensor RdLight1, while simultaneously monitoring or manipulating CIN activity at the same location. CIN stimulation rapidly evoked DA release, and in contrast to slice preparations, this DA release showed no indication of short-term depression or receptor desensitization. The sound of unexpected food delivery evoked a brief joint increase in CIN population activity and DA release, with a second joint increase as rats approached the food. In an operant task, we observed fast ramps in CIN activity during approach behaviors, either to start the trial or to collect rewards. These CIN ramps co-occurred with DA release ramps, without corresponding changes in the firing of lateral VTA DA neurons. Finally, we examined the effects of blocking CIN influence over DA release through local NAc infusion of DHβE, a selective antagonist of β2* nicotinic receptors. DHβE dose-dependently interfered with motivated approach decisions, mimicking the effects of a DA antagonist. Our results support a key influence of CINs over motivated behavior via the local regulation of DA release.

## Editor's evaluation

The manuscript by Mohebi et al. examines the interaction between cholinergic interneurons (CINs) and transmitter release from dopaminergic axons in the nucleus accumbens of behaving animals. The authors propose that CIN activity facilitates dopamine release through activation of DA varicosities, and enhances the motivation to obtain a reward. This study represents an important step forward in our understanding of how dopamine release regulates behavior in concert with acetylcholine signaling, and more broadly how dopamine terminals modulate behavior independently from activity at cell bodies.

## Introduction

Obtaining rewards typically takes work: time-consuming sequences of unrewarded actions before the reward is reached. Deciding when such work is worthwhile is an essential aspect of adaptive behavior.

Dopamine (DA) in the nucleus accumbens (NAc) is a critical modulator of such motivated decision-making (*Berke, 2018*; *Salamone and Correa, 2012*), especially choices to approach potential rewards (*Ikemoto and Panksepp, 1999*; *Nicola, 2010*). Yet how NAc DA release is itself dynamically regulated to achieve appropriate motivation is not well understood.

Some features of DA release mirror the spiking of DA cell bodies. Burst firing (or strong artificial stimulation) of ventral tegmental area (VTA) DA cell bodies is accompanied by a corresponding pulse of NAc DA release (*Mohebi et al., 2019*; *Phillips et al., 2003*; *Suaud-Chagny et al., 1992*). This pulse can encode reward prediction error (RPE), an important learning signal (*Schultz et al., 1997*). Other aspects of DA release appear to be dissociated from spiking. Enhancing motivation by increasing reward availability does not change VTA DA cell firing rates (*Cohen et al., 2012*; *Mohebi et al., 2019*), but does boost NAc Core DA levels, as measured on a time scale of minutes by microdialysis (*Hamid et al., 2016*; *McCullough and Salamone, 1992*). On faster time scales (seconds), many groups have reported ramps in NAc DA as animals approach rewards (*Hamid et al., 2016*; *Howe et al., 2013*; *Roitman et al., 2004*; *Wassum et al., 2012*). These ramps may reflect discounted estimates of future reward (value), a useful motivational signal. Yet we found no evidence for corresponding ramps in spiking rate among identified DA cells in the lateral VTA (*Mohebi et al., 2019*), which provide the major DA input to NAc Core (*Breton et al., 2019*).

An alternative means of controlling DA release involves local circuit components within the NAc (*Sulzer et al., 2016*). Prominent among these are cholinergic interneurons (CINs) which, despite constituting only 1–2% of striatal neurons, provide a very dense network of fibers releasing acetyl-choline (ACh) intermeshed with the DA axon network (*Descarries et al., 1996*). DA axons possess nicotinic ACh receptors (nAChRs) that specifically contain the β2 subunit (β2*; *Jones et al., 2002*). Activation of nAChRs is sufficient to locally evoke action potentials in DA axons, and consequent DA release, even in the absence of DA cell bodies (*Liu et al., 2022*; *Threlfell et al., 2012*; *Cachope and Cheer, 2014*; *Zhou et al., 2001*).

Yet how CINs contribute to NAc DA dynamics in behaving animals is largely unknown. Slice studies suggest that DA release is evoked only by highly synchronous activation of multiple CINs (*Threlfell et al., 2012*), as might occur in response to a salient cue (*Kimura et al., 1984*). Furthermore, DA release in slices shows a lack of summation to repetitive CIN stimulation (*Threlfell et al., 2012*) and strong paired-pulse depression (*Cachope et al., 2012*; *Wang et al., 2014*). This is apparently due to rapid desensitization of nAChRs (*Giniatullin et al., 2005*; *Quick and Lester, 2002*) and feedback inhibition of CINs via DA D2 receptors (*Shin et al., 2017*). These features might limit the ability of CINs to sculpt motivation-related DA release, including ramping. Furthermore, results on CIN contributions to motivation are mixed. Suppression of CINs has been reported to produce a depression-like state in some behavioral tests (*Warner-Schmidt et al., 2012*) but to boost motivation in others (*Collins et al., 2019*).

We examined the relationships between CIN activity, DA release, and motivation. We took advantage of recent advances in optical DA sensors (*Patriarchi et al., 2020*) and a specific operant task in which we previously found a dissociation between DA spiking and release (*Mohebi et al., 2019*). We report that CINs can indeed control DA release in awake behaving animals, but with quite distinct temporal characteristics compared to slices. We find that CIN population activity ramps up during approach behaviors in parallel with DA ramps. Furthermore, selective pharmacological blockade of NAc β2* nAChRs interferes with motivated approach, as does blockade of DA receptors. Our results provide convergent evidence supporting the possibility that CINs are a key mechanism regulating motivation via local control of DA release.

## Results
### ACh reliably drives DA release in NAc, without rapid depression

To study the rapid influence of CINs over striatal DA release in freely moving animals, we employed an all-optical approach. We expressed the excitatory opsin ChR2 in NAc CINs, using ChAT-Cre rats (*Witten et al., 2011*) and Cre-dependent expression from a virus (AAV5::DIO::EF1a::ChR2::eYFP). In the same region, we expressed the red-shifted DA sensor RdLight1 (*Patriarchi et al., 2020*) using another virus (AAV-DJ::CAG::RdLight1). We recorded DA release dynamics by fiber photometry while driving CINs through the same fiber, in awake unrestrained rats not performing any particular task

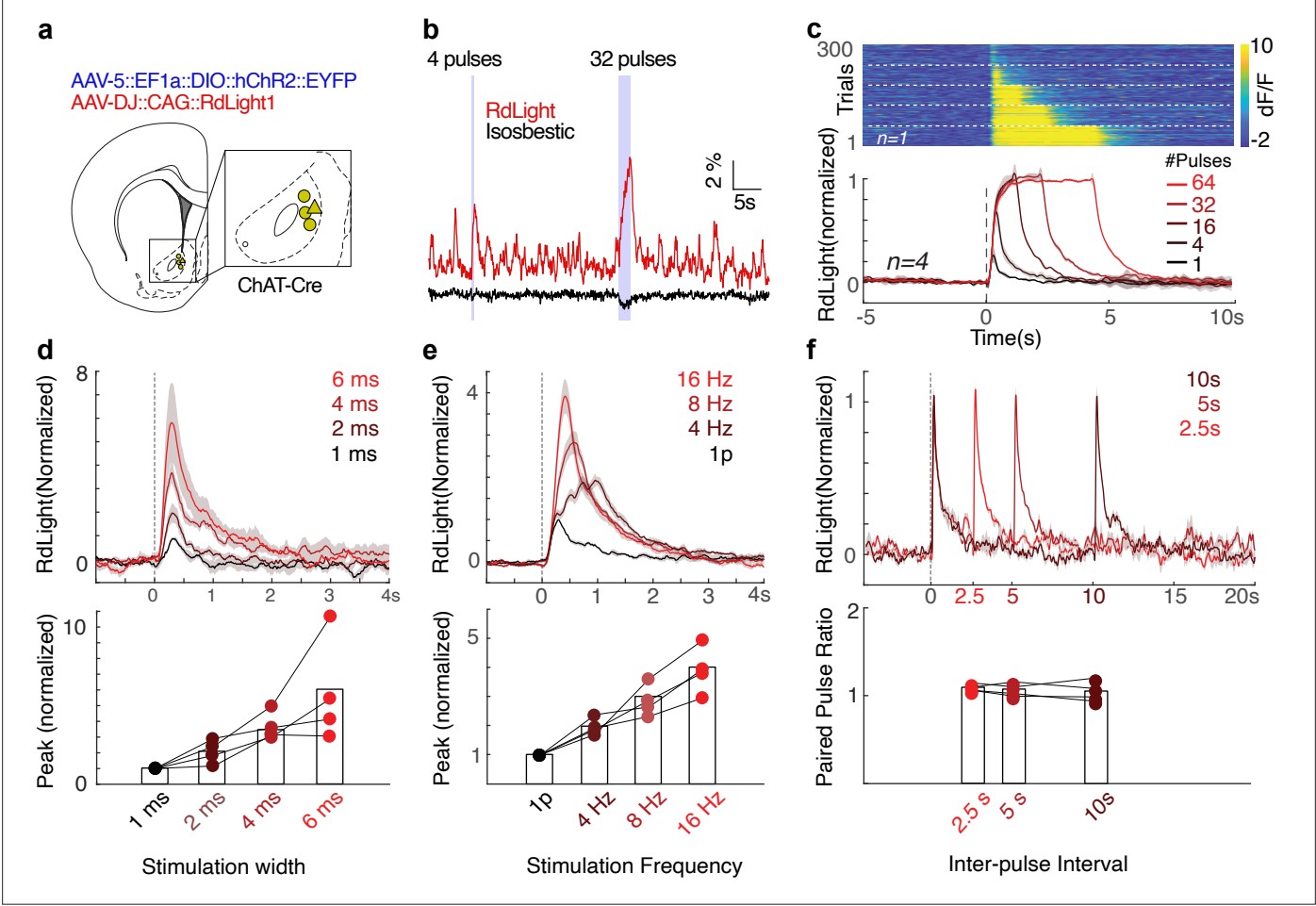

**Figure 1.** Cholinergic interneuron (CIN) stimulation drives dopamine (DA) release in freely moving rats. (**a**) Rat brain atlas section (**Paxinos and Watson, 2006**) showing the approximate location of fiber tips. (**b**) Representative traces of DA release (565 nm excitation, red) and isosbestic control (405 nm, purple), recorded from the location marked by a triangle in (**a**). Laser stimulation was delivered as two trains of pulses (470 nm, 10 mW, 4 ms, 16 Hz). Scale bars: 1 s, 1% dF/F. (**c**) Top: example session showing DA release in response to 16 Hz optogenetic stimulation of nucleus accumbens (NAc) CINs, with 1, 4, 16, 32, or 64 pulses, in pseudo-random order. Dashed lines demarcate trials with similar stimulation parameters. Bottom: normalized dF/F from RdLight aligned to the first pulse in a train of laser stimulation of NAc CINs (Stim parameters: 4 ms pulse, 16 Hz, 10 mW; n = 4 rats). Band shows ± SEM. dF/F traces for each recording were normalized to the maximum response evoked by 64 stimulation pulses. (**d**) DA release in response to a single laser pulse of varying duration (1, 2, 4, 6 ms) at 10 mW. Responses are normalized to 1 ms width evoked response for each subject and averaged. Band shows ± SEM. The magnitude of DA release depends on the pulse width (ANOVA: $F_{(3,12)} = 5.52$, $p=0.013$). (**e**) DA release in response to four laser pulses (4 ms) of varying frequency (4, 8, 16 Hz; all 10 mW). Release patterns are normalized to the single pulse response of the same width and power for each subject, and averaged. Band shows ± SEM. The magnitude of DA release depends on the frequency of stimulation (bottom, ANOVA: $F_{(3,12)} = 9.17$, $p=0.002$). (**f**) Paired-pulse ratio test: DA release patterns in response to a pair of 4 ms, 10 mW pulses. A two-way ANOVA revealed that there was no statistically significant interaction between the pulse order (first or second) and the delay between two pulses ($F_{(2,6)} = 1.05$, $p=0.35$), and there were no main effects of inter-pulse interval ($p=0.47$) or pulse order ($p=0.06$).

The online version of this article includes the following figure supplement(s) for figure 1:

**Figure supplement 1.** No evidence for short-term depression of cholinergic interneuron (CIN)-evoked dopamine (DA) release even at high stimulation frequencies.

(**Figure 1a**). CIN stimulation immediately increased the DA signal, and this increase was maintained as long as stimulation was applied (up to 4 s; **Figure 1b and c**). In contrast to results in slices (**Threlfell et al., 2012**), the amplitude of DA release scaled with the duration and frequency of laser pulses (**Figure 1d and e**). CIN-evoked DA release remained robust to the second of a pair of pulses across a range of inter-pulse intervals (**Figure 1f**), indicating that neither the directly evoked ACh release nor the consequent DA release show rapid synaptic depression in behaving animals (see also **Collins et al., 2019** for related results with choline measurements). To assess short-term plasticity at higher

stimulation frequencies, we conducted an additional quantitative analysis (*Koester and Sakmann, 2000*; ). We compared the observed DA response to four CIN stimulation pulses to that predicted by linearly summing the response to a single pulse, four times. Regardless of stimulation frequency, the ratio between observed and predicted peak response remained close to 1 (*Figure 1—figure supplement 1*), confirming the absence of short-term depression in CIN-evoked DA release.

## CIN activity and DA release each respond to reward-related cues and ramp up during motivated approach

We next examined how CIN activity and DA release co-vary in freely moving rats. We expressed the Ca$^{2+}$ indicator GCaMP6f in CINs (AAV5::Syn::Flex::GCaMP6f) and RdLight1 in the same area (AAV-DJ::CAG::RdLight1). This allowed us to monitor DA release and CIN Ca$^{2+}$ dynamics through the same fiber (*Figure 2a*). CIN activity and DA release showed distinct time courses (*Figure 2b*), but both showed rapid increases in response to the unexpected sound of reward delivery (food hopper 'Click,' occurring at random intervals drawn from a uniform distribution of 15–30 s). After delivery, rats took a variable amount of time to collect the food rewards, allowing us to distinguish between neural changes associated with the sensory cue and those associated with the subsequent food retrieval. Aligning signals on food port entry revealed that both CIN GCaMP and DA release rapidly ramped up as rats approached the food port (*Figure 2c and d*).

## CIN ramps can account for DA release ramps in the absence of DA firing changes

We next turned to a trial-and-error operant task, in which we have previously observed increases in NAc DA release during motivated approach (*Hamid et al., 2016*) without apparent increases in DA cell firing (at least in the lateral VTA; *Mohebi et al., 2019*). In brief, each trial starts with the illumination of a center port (*Figure 3a*). To obtain sucrose pellet rewards, rats approach and place their nose in this center port, and wait for a variable period (500–1500 ms) until an auditory Go cue. They then nose-poke an adjacent port to the left or right. These left/right choices are probabilistically rewarded (10, 50, or 90%, probabilities change independently after blocks of 35–45 trials). Rewarded trials are made apparent by a hopper click, after which rats approach the food port to collect the pellets (*Figure 3b*).

To assess whether ramps in CIN activity might provide the 'missing' control of DA release without changes in DA cell firing, we recorded CIN GCaMP6f signals. We compared these to previously obtained data of VTA DA cell firing and NAc DA release (*Mohebi et al., 2019*). To help disentangle signals related to sensory cue onset versus approach behavior, we examined trials in which light-on and center-in are more than 1 s apart (i.e., 'latency' > 1 s). We measured ramping as the slope of the signal in the last 0.5 s before approach completion. We compared these slopes to a 95% confidence interval generated by measuring slopes at random times during the task (1000 shuffles). None of the optogenetically identified lateral VTA DA cells (0/29) showed significant ramps preceding center-in, but ramps were reliably observed in both DA release (10/10 fiber placements in seven rats) and CIN GCaMP (7/8 fiber placements, five rats). Repeating the same analysis for the food-port-in event produced significant ramping up for none of the 29 DA cells (one ramped down instead), but virtually all recordings of DA release (9/10) and CIN GCaMP (8/8) recordings. These observations are consistent with the hypothesis that ramps in NAc DA release are sculpted via CIN activity rather than DA cell firing.

## Motivated approach relies on NAc β$_2$-containing nicotinic receptors

If NAc CINs directly act upon nearby axons to boost DA release and thereby enhance approach behaviors, such behavior should be sensitive to the blockade of the relevant nAChRs. We tested this in the trial-and-error task, using local drug infusions into the NAc Core (bilaterally, volume 0.5 μl/side). On consecutive days, rats received either aCSF (vehicle), DHβE (selective antagonist of β2* nAChRs [*Changeux, 2018*; *Picciotto et al., 2012*]; 15 or 30 μg/side), or flupenthixol, a non-selective D1 and D2 DA receptor antagonist (FLU; 10 μg/side), 10 min before beginning the task. The drug treatment order was randomly assigned and counterbalanced across rats.

Blockade of β2* nAChRs reduced the number of completed trials in a dose-dependent manner, similarly to (though not as strongly as) FLU (*Figure 4b*). Both FLU and the higher dose of DHβE

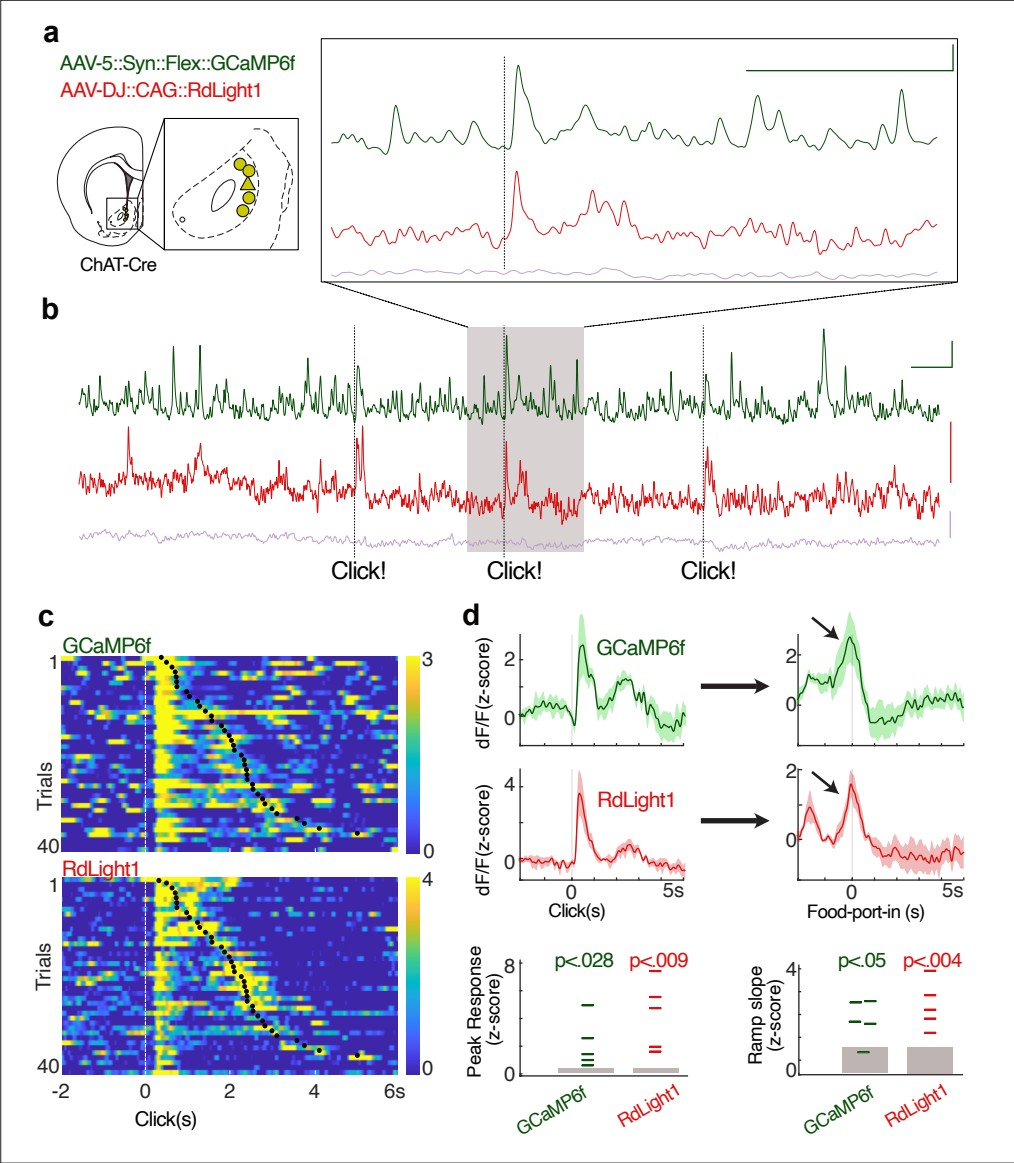

**Figure 2.** Cholinergic interneuron (CIN) activity and dopamine (DA) release similarly respond to reward delivery and ramp up with the reward approach. (**a**) Dual-color photometry of CIN activity and DA fluctuations measured simultaneously through the same fiber. The locations of recording fiber tips are shown projected onto the nearest rat brain atlas section (***Paxinos and Watson, 2006***). (**b**) Example of simultaneous recording of spontaneous activity, from the triangle location in (**a**). Green: CIN GCaMP6f signal (excitation: 470 nm); red: RdLight1 (565 nm); purple: isosbestic control (405 nm). Dashed lines: unexpected food hopper clicks, delivering a sucrose pellet. Inset shows a zoomed-in epoch around a reward click. Note that CIN GCaMP and DA signals are distinct: transients in one signal are not always accompanied by transients in the other. Scale: 1% dF/F, 5 s. (**c**) A representative session with simultaneous CIN GCaMP6f and RdLight1 recording, aligned to hopper clicks. Trials are sorted by the reward collection time (the black dot indicates food-port entry). Both CIN activity and DA fluctuations show a rapid response to the click, and a separate ramping increase during the food-port approach is apparent for longer collection times. Color scale indicates z-scored signal range. (**d**) Top: average traces showing CIN activity and DA fluctuations aligned to the hopper click, peak responses (within 1 s from click) significantly differed from peak values aligned to random time points in the task (n = 1000 shuffles). Bottom: the slope of the signal ramps was significantly different from ramp slopes aligned to random time points throughout the session. The slope was determined by fitting a line that connects the maximum and minimum signal, 0.5 s before the food-port entry. Filled gray rectangles show chance levels (95% confidence ranges, from 1000 shuffles).

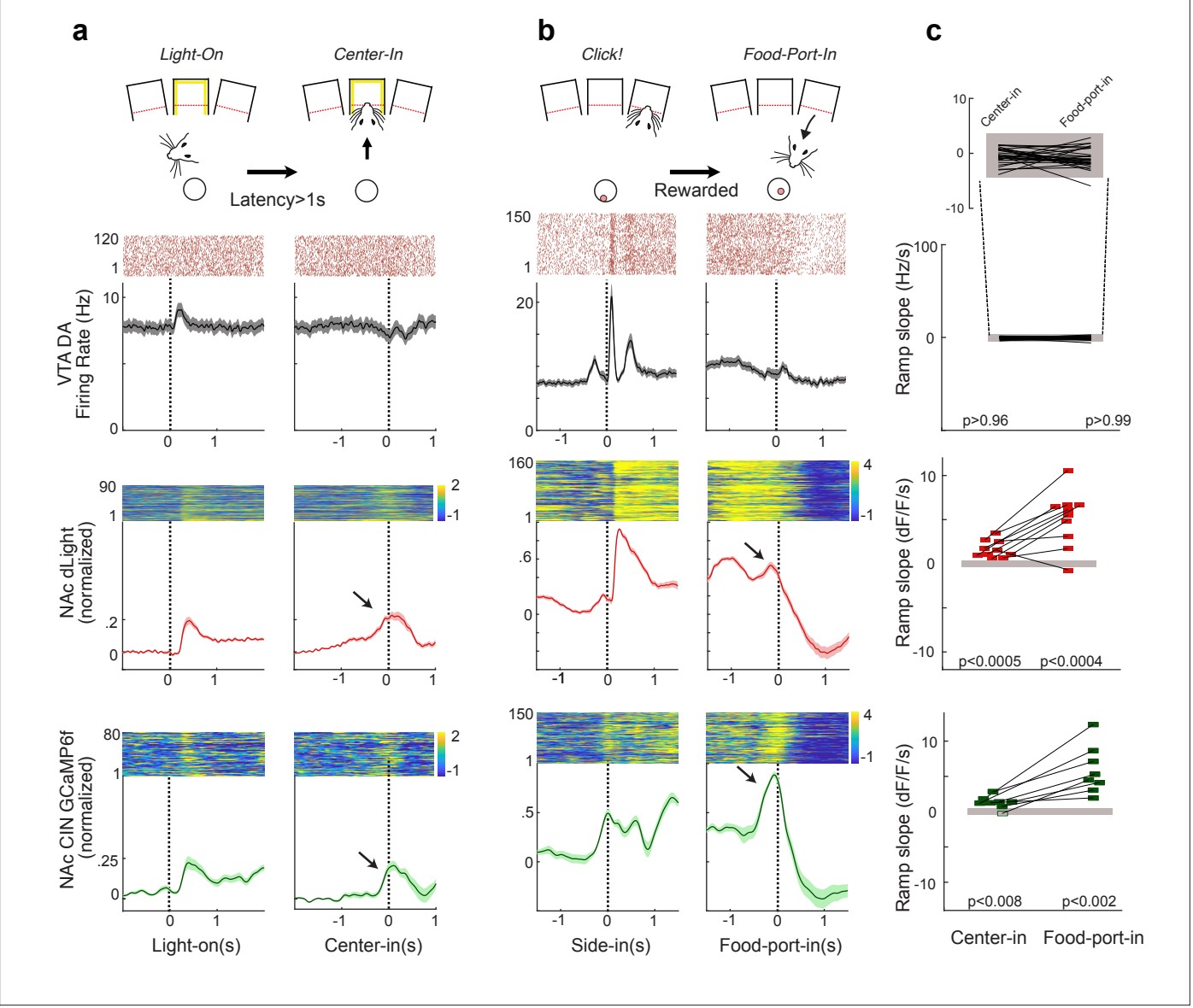

**Figure 3.** In an operant task, cholinergic interneuron (CIN) activity and dopamine (DA) release ramp up without corresponding increases in ventral tegmental area (VTA) DA cell firing. (**a**) Illustration at top: trial initiation involves an approach to an illuminated center port. Data panels: upper and middle data rows: previously reported (*Mohebi et al., 2019*) spiking of identified lateral VTA DA cells (n = 29) and dLight DA signals (n = 10). Lower row shows CIN GCaMP signals (n = 8). Each panel shows a representative single example on top and the population average on the bottom. Shading indicates ± SEM. (**b**) Same as (**a**), but for the approach to the food port following the food delivery Click! sound, on rewarded trials. (**c**) Quantification of slopes during approach. Data points indicate individual DA neurons (top) or fiber placements (middle, bottom). Filled gray rectangles show chance levels (95% confidence ranges, from 1000 shuffles).

increased the latency to initiate a trial (*Figure 4c*). Specifically, both the higher dose of DHβE and FLU decreased the instantaneous likelihood (hazard rate) of decisions to approach the center port, shortly after the center port became illuminated (*Figure 4d*). These observations are consistent with our prior finding that bidirectional optogenetic manipulations of dopamine affect approach hazard rates during the same period (*Hamid et al., 2016*) and support the hypothesis that nAChR-mediated DA release in NAc is involved in the motivation to work.

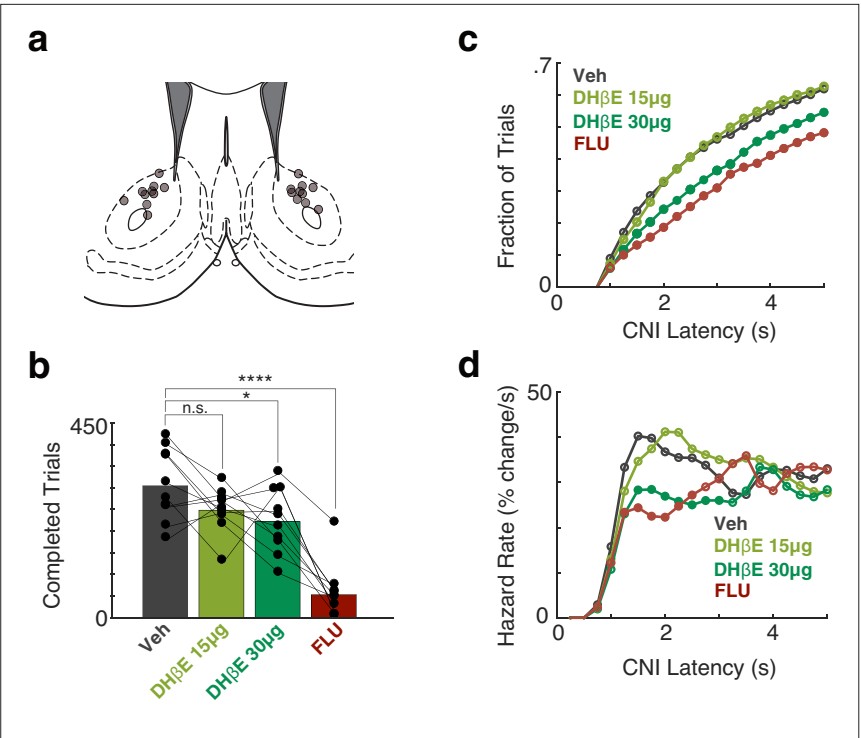

**Figure 4.** Blocking nucleus accumbens (NAc) dopamine or β₂* nicotinic receptors diminishes motivation to work in the operant task. (**a**) Placement of infusion cannula tips (circles) in NAc Core, based on postmortem histology. (**b**). The number of trials completed during a 2 hr Bandit session compared to the veh session. $F(3,36) = 24.64$, $p<7.6 \times 10^{-9}$. Rats treated with DHβE-30 μg and FLU completed fewer trials compared to the vehicle session ($p<0.013$, $p<1 \times 10^{-9}$, respectively). DHβE-15μg treatment caused a trend toward fewer completed trials ($p=0.074$) (**c**). Cumulative distribution of long latencies (latency >1 s) for different treatment conditions, binned at 250 ms. Filled circles denote statistical differences from the Vehicle condition ($p<0.01$). (**d**) The hazard rate of long latencies for different condition treatments demonstrates that blocking nicotinic transmission decreases the likelihood of a self-initiated approach (between 1 and 2 s from light-on) in a dose-dependent manner. Filled circles denote statistical differences from the veh condition ($p<0.01$).

## Discussion

Local regulation of striatal DA by ACh has long been established in brain slices (*Giorguieff et al., 1976*). However, how this influence shapes DA signals in vivo has remained obscure. As a result, cholinergic modulation of DA release has not typically been incorporated into functional accounts of DA dynamics (*Berke, 2018*), which have instead emphasized the faithful transmission of RPEs encoded in DA cell spiking. An accurate characterization of local modulation of NAc DA in behaving animals is essential not simply for understanding the neural control of motivation, but also altered physiological states evoked by disorders or abused drugs (notably, nicotine).

Seminal slice studies reported that while CINs can profoundly influence DA release via β2* nicotinic receptors, this effect shows strong short-term depression that can take minutes to recover (*Cachope et al., 2012*; *Threlfell et al., 2012*). If these desensitizing mechanisms apply in the awake behaving case, we might expect little functional contribution of CINs to DA signaling, and that stimulating CINs would not evoke appreciable extra DA release. Our results clearly demonstrate otherwise: stimulating CINs evoked strong DA release in a frequency-dependent manner, with no evidence of depression. This result both confirms the presence of powerful ACh modulation of DA release under physiological conditions and demonstrates that this modulation has the dynamic range to be a major, continuing influence over behaviorally relevant DA signals.

What can explain the discrepancy between ex vivo and in vivo experiments? One possibility involves the very different conditions for acetylcholinesterase (AChE), an enzyme that provides critical constraints over ACh signaling dynamics. AChE functions very rapidly, almost approaching the rate of ACh diffusion (*Dvir et al., 2010*; *Quinn, 1987*), but this rate will be slower at the lower temperature

of slice experiments compared to whole animals. Furthermore, AChE is soluble in the extracellular matrix (*Sirviö et al., 1989*), and so can be partly washed away in the slice preparation. Taken together, released ACh in the slice may escape degradation in the synaptic space for longer than normal, and thereby provoke desensitization of a larger proportion of nAChRs.

One of the most prominent reported features of striatal CINs is a reliable pause in response to reward-predicting cues, in contrast to DA release (*Morris et al., 2004*). However, at least in NAc we found that both DA release and CIN activity increase rapidly in response to such cues (see also *Atallah et al., 2014*; *Howe et al., 2019*). Of course, bulk CIN GCaMP signals are at best an incomplete readout of either CIN firing or the spatiotemporal dynamics of ACh release. In particular, imaging of CINs GCaMP in the dorsal striatum found a mismatch between signals at cell bodies and the rest of the field of view (considered to be neuropil; *Rehani et al., 2019*). Understanding whether control of DA release via nAChRs directly reflects CIN firing will require more experiments adding either opto-genetic tagging of CINs (*Duhne, 2022*) or high-resolution voltage imaging (*Piatkevich et al., 2019*). One related important direction for future studies is to understand why some transient fluctuations in CIN activity evoke DA release (*Liu et al., 2022*) but others do not (e.g., *Figure 2b*).

In animals performing an operant task, we demonstrated a correspondence between increasing ramps in NAc DA release during motivated approach and increasing NAc CIN activity (measured with GCaMP), at times when VTA DA cell firing is not increasing. This observation supports the idea that CINs can drive behaviorally relevant DA release even in the absence of DA firing changes. We further showed that, in the same task, interfering with NAc β2* nAChRs reduces the likelihood that rats perform the motivated approach behavior. However, one limitation of this study is that we did not causally establish that CINs are *directly* driving DA release, via β2* nAChRs, at the *specific* moments of DA ramping during approach. In subsequent studies, we hope to do this through combined local pharmacology, optogenetics, and photometry in behaving animals, as technical advances permit. Designing a conclusive manipulation experiment is not trivial; for example, if CINs tonically promote DA release, simple suppression of CINs could interfere with DA ramps during approach, even if CINs do not normally drive ramping.

Given that ACh can promote NAc DA release and thereby motivation, a continuing conundrum is to explain long-standing evidence for *opposing* roles for striatal DA and ACh in behavioral control (*Mcgeer et al., 1961*; *Hikida et al., 2001*; *Collins et al., 2019*). One potentially relevant behavioral dimension is whether actions are prompted by external events versus being 'self-initiated.' Increases in both (presumed) CIN activity and striatal DA release have been reported in conjunction with self-timed actions (e.g., *Lee et al., 2006*; *Hamilos et al., 2021*). This seems consistent with our observa-tion of prominent joint DA and CIN activity ramps in longer-latency trials in our operant task, where rats are not simply reacting to the trial start light, but rather approaching at a variable time after its onset. However, if CIN control of NAc DA release is always important for self-initiated, motivated behavior, DHβE should suppress the hazard rate of approach across a wide range of delays after the light onset. In fact, both DHβE and FLU particularly depressed approach behavior in the 1–3 s range after light onset, with the hazard rate returning to control levels by ~4 s (*Figure 4d*). This suggests that the NAc joint ACh-DA mechanisms investigated here may be particularly involved in deciding whether it is worthwhile reorganizing ongoing behavior to respond to an external cue (*Blazquez et al., 2002*).

Finally, if CINs do indeed help locally sculpt motivational aspects of DA release, a natural question becomes how relevant CIN signals are themselves generated. CINs receive inputs from a wide range of areas that can supply motivational information and influence DA release, including the basolateral amygdala (*Floresco et al., 1998*), parafascicular thalamus (*Bradfield et al., 2013*), frontal cortex (*Kosillo et al., 2016*), and long-range GABAergic subpopulations in the VTA (*Al-Hasani et al., 2021*; *Brown et al., 2012*). They also sample the state of surrounding networks of striatal projection cells, allowing them to potentially extract useful decision variables (*Franklin and Frank, 2015*). Resolving how motivational signals are computed remains an ongoing challenge, but monitoring and manip-ulating each of these various circuit components within the same behavioral context is a promising approach.

## Methods

**Key resources table**

| Reagent type (species) or resource | Designation | Source or reference | Identifiers | Additional information |
|---|---|---|---|---|
| Strain, strain background (LE rats, both male and female) | ChAT-Cre, Long-Evans (male and female rats) | RRRC | Long-Evans-Tg(ChAT-Cre)5.1Deis | |
| Recombinant DNA reagent | AAV5-EF1a-DIO- ChR2-eYFP | Addgene | Cat# 20298-AAV5 | |
| Recombinant DNA reagent | pAAV.Syn.Flex.GCaMP6f.WPRE.SV40 | Addgene | Cat# 100833-AAV5 | |
| Recombinant DNA reagent | AAV-CAG-RdLight1 | *Patriarchi et al., 2020* | | |
| Chemical compound, drug | DHβE | Tocris Bioscience | Cat# 2349 | |
| Chemical compound, drug | Flupenthixol | Sigma-Aldrich | CAS# 51529-01-2 | |
| Software, algorithm | MATLAB | MathWorks, Inc. | | |
| Software, algorithm | LabView | National Instruments | | |

### Animals

In the conduct of this study, the animals involved were accommodated at the University of California San Francisco (UCSF) Animal Research Facility, which is accredited by AAALAC (#001084). The facility strictly adheres to institutional, federal, and AAALAC guidelines to ensure the highest standards of animal care. Procedures such as euthanasia for perfusion fixation were performed under profound anesthesia to minimize discomfort. The use of animals in this study was in strict compliance with the Public Health Service Policy on Humane Care and Use of Laboratory Animals. The UCSF Institutional Animal Care and Use Committee granted approval for this study (protocol # AN196232-01B). Furthermore, UCSF holds a PHS-approved Animal Welfare Assurance D16-00253/A3400-01, further affirming our commitment to ethical animal use.

### Behavioral task

Rats were trained in the operant 'bandit' task described in detail previously (*Hamid et al., 2016*; *Mohebi et al., 2019*). Reward probabilities for left and right choices were pseudorandomly chosen from 10, 50, or 90% for blocks of 35–45 trials, over an ~2 hr session. Procedural errors (e.g., failure to hold until the Go cue) resulted in illumination of the house light and a timeout period of 1.5 s.

### Photometry and optogenetic stimulation

We used a viral approach to express the genetically encoded optical DA sensor RdLight1 and the $Ca^{2+}$ indicator GCaMP6f. Under isoflurane anesthesia, a 1 µl cocktail of AAV-DJ-CAG-RdLight1 and AAV5-Syn-Flex-GCaMP6f was slowly (100 nl/min) injected (Nanoject III, Drummond) through a glass micropipette targeting nucleus accumbens: (AP: 1.7, ML: 1.7, DV: 7.0 mm relative to bregma). During the same surgery, optical fibers (400 µm core, 430 µm total diameter) attached to a metal ferrule (Doric) were inserted (target depth 200 µm higher than AAV) and cemented in place. Data were collected >3 wk later, to allow for transgene expression. We used time-division multiplexing (*Akam and Walton, 2019*): green (565 nm), blue (470 nm), and violet (405 nm; isosbestic control) LEDs were alternately switched on and off in 10 ms frames (4 ms on, 6 ms off). Excitation power at the fiber tip was set to 30 µW for each wavelength. Both excitation and emission signals passed through minicube filters (Doric), and bulk fluorescence was measured with a femtowatt detector (Newport, Model 2151) sampling at 10 kHz. The separate signals were then rescaled to each other via a least-square fit before analysis.

To combine fiber photometry with optogenetics, we infused a cocktail of AAV-DJ-CAG-RdLight1 and AAV5-EF1a-DIO-ChR2-eYFP into the nucleus accumbens and implanted 400 µm fibers above the infusion site during the same surgery. We used time division multiplexing with interleaved LED illumination (565 nm, 405 nm), as above. We used brief laser pulses (470 nm, 20 mW) to activate ChR2.

### Drugs and infusion procedures

Rats were bilaterally infused with either the selective α4β2 nicotinic antagonist DHβE (low dose: 15 µg/0.5 µl/side, high dose 30 µg/0.5 µl/side; Tocris Bioscience, Bristol, UK) the non-selective

dopamine receptor antagonist flupenthixol (15 μg/0.5 μl/side; Tocris) or sterile saline (0.5 μl/side), which served as the vehicle control. All rats received all drug infusions, counterbalanced for order. Drugs were infused as described previously (*Collins et al., 2016*). Briefly, dummies were removed, and injectors were inserted into the guide cannulae protruding 3 mm past the guide tip targeted to the NAc. Using a micro infusion pump (Harvard Apparatus), drugs were infused at a rate of 0.5 μl/min for 1 min. Following a 1 min post-infusion wait time, injectors were removed and dummies inserted. Rats were given 10 min post-infusion before being placed in operant chambers for the bandit task.

## Additional information

### Funding

| Funder | Grant reference number | Author |
| --- | --- | --- |
| National Institute on Drug Abuse | R01DA045783 | Joshua D Berke |
| National Institute of Neurological Disorders and Stroke | R01NS123516 | Joshua D Berke |
| National Institute of Neurological Disorders and Stroke | R01NS116626 | Joshua D Berke |
| National Institute on Alcohol Abuse and Alcoholism | R21AA027157 | Joshua D Berke |
| National Institute of Mental Health | K01MH126223 | Ali Mohebi |
| Brain and Behavior Research Foundation | NARSAD YIA 29361 | Ali Mohebi |

The funders had no role in study design, data collection and interpretation, or the decision to submit the work for publication.

### Author contributions

Ali Mohebi, Conceptualization, Data curation, Software, Formal analysis, Funding acquisition, Investigation, Visualization, Methodology, Writing - original draft, Writing - review and editing; Val L Collins, Investigation, Methodology; Joshua D Berke, Conceptualization, Resources, Data curation, Supervision, Funding acquisition, Investigation, Visualization, Project administration, Writing - review and editing

### Author ORCIDs

Ali Mohebi ⬩ http://orcid.org/0000-0001-7291-3448
Joshua D Berke ⬩ http://orcid.org/0000-0003-1436-6823

### Ethics

In the conduct of this study, the animals involved were accommodated at the University of California San Francisco (UCSF) Animal Research Facility, which is accredited by AAALAC (#001084). The facility strictly adheres to institutional, federal, and AAALAC guidelines to ensure the highest standards of animal care. Procedures such as euthanasia for perfusion fixation were performed under profound anesthesia to minimize discomfort. The use of animals in this study was in strict compliance with the Public Health Service Policy on Humane Care and Use of Laboratory Animals. The UCSF Institutional Animal Care and Use Committee granted approval for this study (protocol # AN196232-01B). Furthermore, UCSF holds a PHS-approved Animal Welfare Assurance D16-00253/A3400-01, further affirming our commitment to ethical animal use.

### Decision letter and Author response
Decision letter https://doi.org/10.7554/eLife.85011.sa1
Author response https://doi.org/10.7554/eLife.85011.sa2

## Additional files

### Supplementary files
• MDAR checklist

### Data availability
All data generated or analyzed during this study will be made publicly available on Dryad (http://doi.org/10.7272/dryad.Q68P5XST).

The following dataset was generated:

| Author(s) | Year | Dataset title | Dataset URL | Database and Identifier |
|---|---|---|---|---|
| Mohebi A, Collins VL, Berke JD | 2022 | Cholinergic Interneurons and Dopamine in the Nucleus Accumbens | http://doi.org/10.7272/dryad.Q68P5XST | Dryad Digital Repository, 10.7272/dryad.Q68P5XST |

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
