## [Editor Report]

The manuscript by Mohebi et al. examines the interaction between cholinergic interneurons (CINs) and transmitter release from dopaminergic axons in the nucleus accumbens of behaving animals. The authors propose that CIN activity facilitates dopamine release through activation of DA varicosities, and enhances the motivation to obtain a reward. This study represents an important step forward in our understanding of how dopamine release regulates behavior in concert with acetylcholine signaling, and more broadly how dopamine terminals modulate behavior independently from activity at cell bodies.

---

## [Decision Letter]

**Decision letter after peer review:**

Thank you for submitting your article "Cholinergic Interneurons Drive Motivation by Promoting Dopamine Release in the Nucleus Accumbens" for consideration by *eLife*. Your article has been reviewed by 3 peer reviewers, and the evaluation has been overseen by a Reviewing Editor and Michael Taffe as the Senior Editor. The reviewers have opted to remain anonymous.

Essential revisions:

1) Although there is significance to the paper, all reviewers are of the mindset that the authors need to show that, at physiologically relevant patterns of activity, CINs drive DA release in a behaviorally-relevant manner (manipulating CIN activity while at the same time measuring behaviorally evoked DA fluorescence, for example). This experiment is crucial in light of recent papers showing a disconnect between the two processes in behaving animals (the recent work of the Sabatini and Tritsch labs).

2) It is also very important for the authors to show that optically evoked DA fluorescence is indeed dependent on a4b2 receptors (see Mateo et al., 2017).

3) Please thoroughly explain the observation of a lack of frequency-dependent depression, which greatly contrasts with previously published reports (Cachope et al., 2012; Threlfell et al., 2012).

4) Please clarify whether independent single-unit VTA DA neuron data was collected for this study.

*Reviewer #1 (Recommendations for the authors):*

Detailed comments follow below.

– Figure 1 shows that synchronous optogenetic activation of CINs induces DA release in vivo. These results confirm those obtained in previous slice experiments, although the lack of frequency-dependent depression is intriguing. The authors discuss potential reasons such as the different ACH-E dynamics in slices vs in vivo, temperature effects, and more. Could this difference also be due to the methods used to measure DA release (voltammetry vs optical)?

– The dual wavelength imaging presented in figures 2-3 shows a high similarity between the DA signal and CIN activity in response to auditory stimuli and approach to the reward port. Are the simultaneous measurements completely independent? No chance of "cross talk" between the channels? Would the results hold if the imaging of CIN calcium activity and DA release were measured separately in different animals, or simultaneously but in different hemispheres?

– It is also not completely clear if the spiking data was obtained simultaneously or if it is actually the data from a previous paper by the lab. It says in the figure legend that the spiking activity of DA neurons during this task was previously reported, but does this mean that only the imaging data was acquired simultaneously while the spiking data are from different animals?

– Figure 3 indeed shows that the ramping in DA release is not accompanied by ramping in VTA cell spiking but with a ramping of CIN activity. While this does not contradict the potential involvement of CINs in mediating the DA ramp, it does not prove it.

– One open question regarding the local DA release vs spiking activity is whether CIN-induced release generates spikes that backpropagate to the cell bodies. Using the experimental setup presented in figure 3, the authors are positioned to answer this question by optogenetically activating CINs and recording backpropagating spikes in VTA cells.

– The last figure (4) shows a decrease in performance in the operant task following the application of DHBE in the NAc. The impact of the DA agonist FLU was clearly larger, and the DHBE application was effective only in higher concentrations. The changes in behavioral performance are clear in both FLU and DHBE-30ug applications. This does not, however, prove that the impact of DHBE is mediated via DA release. Such a strong manipulation of nicotinic receptor-mediated transmission would affect other cells in the NAc, including several types of GABAergic interneurons (See papers by Assous 2021, 2022). It is therefore not clear that the data presented in Fig, 4 adds to our understanding of CIN-mediated DA release.

– This leads me to the main concern, that the role of CINs in mediating DA release is not directly shown. It is shown that synchronous optogenetic activation of CINs indeed causes DA release (Figure 1). It is also shown that CIN and DA release are often correlated, such as in approach-related ramping and sensory responses. However, whether CIN-induced DA release underlies these correlated activities is not proven. This could be done by suppressing CIN activity and showing the impact on DA release. Will the ramping be affected? Will the sensory responses decrease? Manipulation of CIN activity can be done by optogenetic or chemogenetic inhibition while imaging DA release.

– Two recent studies by the Sabatini and Tritsch labs address similar questions regarding the interactions between CINs and DA release. Although the striatal regions tested are not the same as in this study, a common claim is that there is little impact of CINs on DA release in behaving mice and that to a large degree their fluctuations are negatively correlated. It would be important to discuss the discrepancies between these recent studies, in addition to the comparison with older slice studies.

*Reviewer #2 (Recommendations for the authors):*

– The data presented here suggests that "ramp" increases in spiking do not occur. However, this stands in contrast to data from the Witten lab showing that ramping does occur in somatic spiking. Could a difference in recording region or experimental approach account for this difference?

– The authors mention in the discussion that "stimulating CINs evoked strong DA release…". Strong relative to what? The signals in this study are reported as normalized values or presented as z-scores in nearly all instances. The authors should provide clearer justification based on absolute data values for why they think the CIN-evoked DA signals are strong.

– In the final paragraph of the results, flupenthixol is probably more accurately described as a "non-selective dopamine receptor antagonist."

– It would be helpful to give readers an idea of where electrical VTA DA neurons were made (lateral, medial, etc).

[Editors' note: further revisions were suggested prior to acceptance, as described below.]

Thank you for resubmitting your work entitled "Cholinergic Interneurons Drive Motivation by Promoting Dopamine Release in the Nucleus Accumbens" for further consideration by *eLife*. Your revised article has been evaluated by Michael Taffe (Senior Editor) and a Reviewing Editor.

The manuscript has been improved but there are some remaining issues that need to be addressed, as outlined below:

The reviewers find that your rebuttal is fine on all points, except for the key experiment showing that cholinergic interneurons causally drive motivation by promoting dopamine release. In light of the fact that you could not complete the required experiments, this needs to be thoroughly addressed as a limitation in the discussion. All reviewers suggest toning down the message in the title and in the introduction (lines 86-88), to claim that the results support the possibility of CINs sculpting DA release, as this is not directly and unambiguously demonstrated.

---

## [Author Response]

Essential revisions:Reviewer #1 (Recommendations for the authors):Detailed comments follow below.– Figure 1 shows that synchronous optogenetic activation of CINs induces DA release in vivo. These results confirm those obtained in previous slice experiments, although the lack of frequency-dependent depression is intriguing. The authors discuss potential reasons such as the different ACH-E dynamics in slices vs in vivo, temperature effects, and more. Could this difference also be due to the methods used to measure DA release (voltammetry vs optical)?

We did not perform a direct comparison of voltammetry vs optical methods in this manuscript, but based on prior work by others and ourselves we think this is unlikely to be a critical factor. When used appropriately both methods are capable of following sub-second DA dynamics, and have sufficient dynamic range to examine both facilitating and depressing effects. As just one example, Rice and Cragg (2004) used voltammetry in slices and observed short-term depression of DA release to multiple electrical pulses. However this depression was not an artifact of the recording method, as it was alleviated in the presence of nicotine. In our in vivo photometry experiments we carefully selected stimulation parameters to evoke transients that are comparable to naturally occurring DA transients, and we are not aware of any reason to think that we would have not seen depression of this response if it were present.

– The dual wavelength imaging presented in figures 2-3 shows a high similarity between the DA signal and CIN activity in response to auditory stimuli and approach to the reward port. Are the simultaneous measurements completely independent? No chance of "cross talk" between the channels? Would the results hold if the imaging of CIN calcium activity and DA release were measured separately in different animals, or simultaneously but in different hemispheres?

It is indeed important to consider cross-talk, but we can rule it out because:

1) In a previous study (Patriarchi, Mohebi et al., Nature Methods 2020), we demonstrated that cross-talk between RdLight1 and GCaMP6f is negligible.

2) In simultaneous measurements we observe clear examples of transients in CIN activity that do not correspond to DA transients, and vice versa (examples in Figure 2 of the current manuscript)

3) The current manuscript does include both simultaneous measurements (Figure 2), and recordings in separate animals (Figure 3). In each case we observe concurrent ramps in CIN GCaMP6f and RdLight1 signals during motivated approach behaviors. In contrast, at other times (reward delivery at Side-In, Figure 3b) we see a large increase in DA release *without* a corresponding large change in CIN GCaMP signal. DA release at that time instead mirrors a large increase in DA cell firing, consistent with the idea that DA release can be controlled in multiple ways in behaving animals.

– It is also not completely clear if the spiking data was obtained simultaneously or if it is actually the data from a previous paper by the lab. It says in the figure legend that the spiking activity of DA neurons during this task was previously reported, but does this mean that only the imaging data was acquired simultaneously while the spiking data are from different animals?

We have now clarified in the text that for the operant task (Figure 3), we used three separate cohorts of rats to record (1) spiking of DA cells, (2) photometry of dLight, and (3) photometry of CIN activity, and clarified that data from the first two cohorts was previously reported in Mohebi et al. 2019.

– Figure 3 indeed shows that the ramping in DA release is not accompanied by ramping in VTA cell spiking but with a ramping of CIN activity. While this does not contradict the potential involvement of CINs in mediating the DA ramp, it does not prove it.

We acknowledge that our current study has significant limitations in definitively establishing that CIN activity drives behavior-linked DA ramps. In principle, a helpful experiment would be to simultaneously monitor DA release while antagonizing the nAChRs on DA terminals, during behaviors when DA ramps normally occur. Such an experiment is not within our current technical capabilities. Furthermore, even this experiment might not be conclusive: for example, if there is “baseline” DA release evoked by CINs, then suppressing CINs could interfere with observed ramping even if CINs do not normally drive ramping. For these reasons we believe it is appropriate to proceed with the current data set, while acknowledging its limitations (which we do in the revised Discussion).

– One open question regarding the local DA release vs spiking activity is whether CIN-induced release generates spikes that backpropagate to the cell bodies. Using the experimental setup presented in figure 3, the authors are positioned to answer this question by optogenetically activating CINs and recording backpropagating spikes in VTA cells.

Whether CIN-induced DA release causes DA spikes that backpropagate all the way back to cell bodies is a very interesting question, but it is beyond the scope of the current manuscript. To date we have not performed experiments in which we record DA cell spiking while optogenetically activating CINs.

– The last figure (4) shows a decrease in performance in the operant task following the application of DHBE in the NAc. The impact of the DA agonist FLU was clearly larger, and the DHBE application was effective only in higher concentrations. The changes in behavioral performance are clear in both FLU and DHBE-30ug applications. This does not, however, prove that the impact of DHBE is mediated via DA release. Such a strong manipulation of nicotinic receptor-mediated transmission would affect other cells in the NAc, including several types of GABAergic interneurons (See papers by Assous 2021, 2022). It is therefore not clear that the data presented in Fig, 4 adds to our understanding of CIN-mediated DA release.

If we had observed that DHBE infusion into NAc does *not* affect initiation of trials in our instrumental task, this would have contradicted our hypothesis that CIN control over DA release via nAChRs contributes to motivation to work. Observing that we *do* see such an effect is therefore useful support for our hypothesis. However we agree that it is certainly possible that more indirect effects are involved instead, and we have updated the Discussion section to reflect this point.

– This leads me to the main concern, that the role of CINs in mediating DA release is not directly shown. It is shown that synchronous optogenetic activation of CINs indeed causes DA release (Figure 1). It is also shown that CIN and DA release are often correlated, such as in approach-related ramping and sensory responses. However, whether CIN-induced DA release underlies these correlated activities is not proven. This could be done by suppressing CIN activity and showing the impact on DA release. Will the ramping be affected? Will the sensory responses decrease? Manipulation of CIN activity can be done by optogenetic or chemogenetic inhibition while imaging DA release.

As noted above, our study has limitations that we are now careful to acknowledge in the Discussion.

– Two recent studies by the Sabatini and Tritsch labs address similar questions regarding the interactions between CINs and DA release. Although the striatal regions tested are not the same as in this study, a common claim is that there is little impact of CINs on DA release in behaving mice and that to a large degree their fluctuations are negatively correlated. It would be important to discuss the discrepancies between these recent studies, in addition to the comparison with older slice studies.

We have also noted those pre-prints with interest, even though they investigated different brain regions using different approaches. There are established differences between CIN-DA interactions in dorsal vs. ventral striatum that we suspect are relevant here. But given the rapid pace of developments in this subfield, we prefer not to speculate too much at this point and instead review the overall body of work once it is published.

Reviewer #2 (Recommendations for the authors):– The data presented here suggests that "ramp" increases in spiking do not occur. However, this stands in contrast to data from the Witten lab showing that ramping does occur in somatic spiking. Could a difference in recording region or experimental approach account for this difference?

These are both possibilities. In particular we note that the ramps reported by the Witten lab are actually ramps in the calcium activity of DA cells, monitored using GCaMP6, which has different dynamics and may signal distinct processes compared to spiking.

– The authors mention in the discussion that "stimulating CINs evoked strong DA release…". Strong relative to what? The signals in this study are reported as normalized values or presented as z-scores in nearly all instances. The authors should provide clearer justification based on absolute data values for why they think the CIN-evoked DA signals are strong.

By "strong," we simply mean that the CIN-evoked DA signals are larger than most spontaneous fluctuations (Figure 1b), and comparable in signal:noise ratio to the large transients evoked by natural reward cues.

– In the final paragraph of the results, flupenthixol is probably more accurately described as a "non-selective dopamine receptor antagonist."

Agreed. Given this comment and that of Reviewer 3 we have now modified the text to say "a non-selective D1 and D2 DA receptor antagonist".

– It would be helpful to give readers an idea of where electrical VTA DA neurons were made (lateral, medial, etc).

We have now modified the text to specify lateral VTA.

[Editors' note: further revisions were suggested prior to acceptance, as described below.]

The manuscript has been improved but there are some remaining issues that need to be addressed, as outlined below:The reviewers find that your rebuttal is fine on all points, except for the key experiment showing that cholinergic interneurons causally drive motivation by promoting dopamine release. In light of the fact that you could not complete the required experiments, this needs to be thoroughly addressed as a limitation in the discussion. All reviewers suggest toning down the message in the title and in the introduction (lines 86-88), to claim that the results support the possibility of CINs sculpting DA release, as this is not directly and unambiguously demonstrated.

In response to the issue you mentioned we have:

– Amended the Title, to make the claim about the relationship between CIN-evoked DA release and the control of motivation via nicotinic receptors less strong.

– Amended the Introduction. In the specific place you flagged we now state that our results merely support the possibility that CINs regulate motivation via local control of DA.

– Amended the Discussion. We now have both an explicit statement describing the limitation of this experiment in not combining causal manipulations together with DA sensing, together with a line noting that designing an interpretable experiment along these lines is not as simple as it might seem.